# Amorphous Solid Dispersion of Hesperidin with Polymer Excipients for Enhanced Apparent Solubility as a More Effective Approach to the Treatment of Civilization Diseases

**DOI:** 10.3390/ijms232315198

**Published:** 2022-12-02

**Authors:** Natalia Rosiak, Kamil Wdowiak, Ewa Tykarska, Judyta Cielecka-Piontek

**Affiliations:** 1Department of Pharmacognosy, Faculty of Pharmacy, Poznan University of Medical Sciences, 3 Rokietnicka St., 60-806 Poznan, Poland; 2Department of Chemical Technology of Drugs, Poznan University of Medical Sciences, 6 Grunwaldzka St., 60-780 Poznan, Poland

**Keywords:** hesperidin, amorphous solid dispersion, apparent solubility, glass transition, Gordon–Taylor equation, DFT calculation

## Abstract

The present study reports amorphous solid dispersions (ASDs) of hesperidin (Hes) prepared by ball milling to improve its solubility and apparent solubility over the unmodified compound. The carriers were Soluplus^®^ (Sol), alginate sodium (SA), and hydroxypropylmethylcellulose (HPMC). XRPD analysis confirmed full amorphization of all binary systems in 1:5 *w/w* ratio. One glass transition (T_g_) observed in DSC thermograms of hesperidin:Soluplus^®^ (Hes:Sol) and hesperidin:HPMC (Hes:HPMC) 1:5 *w/w* systems confirmed complete miscibility. The mathematical model (Gordon–Taylor equation) indicates that the obtained amorphous systems are characterized by weak interactions. The FT-IR results confirmed that hydrogen bonds are responsible for stabilizing the amorphous state of Hes. Stability studies indicate that the strength of these bonds is insufficient to maintain the amorphous state of Hes under stress conditions (25 °C and 60 °C 76.4% RH). HPLC analysis suggested that the absence of degradation products indicates safe hesperidin delivery systems. The solubility and apparent solubility were increased in all media (water, phosphate buffer pH 6.8 and HCl (0.1 N)) compared to the pure compound. Our study showed that all obtained ASDs are promising systems for Hes delivery, wherein Hes:Sol 1:5 *w/w* has the best solubility (about 300-fold in each media) and apparent solubility (about 70% in phosphate buffer pH 6.8 and 63% in HCl).

## 1. Introduction

The health-promoting potential of polyphenols has been proven in a number of in vitro and in vivo studies [1,2,3,4]. However, expanding knowledge in this area increasingly raises the issue of the bioavailability of polyphenols and the factors that can modify them. The main factor determining natural compounds’ activity is their solubility [1,5]. Hesperidin (Hes) is an example of a polyphenol that occurs in citrus fruit (sweet orange, lemon). This flavanone glycoside has antioxidant, anti-inflammatory, antimicrobial, anticarcinogenic, and antiallergic activity [6]. Hesperidin therapy is considered GRAS (Generally recognized as safe) (website: https://www.fda.gov/media/133322/download, accessed on 19 November 2022), non-accumulative, and has been found to be nontoxic even in pregnancy [7]. In addition, 51 animal studies by Li et al. confirmed that hesperidin isolated from citrus fruit showed a good safety profile [8]. Next, based on an oral preclinical toxicity study, Cicero-Sarmiento et al. suggest that their developed naringenin–hesperidin mixtures are potential safety drug candidates for clinical studies [9]. Hes plays a protective role against metabolic syndrome and associated diseases. Numerous clinical studies indicate that hesperidin as a pure compound can be used in patients with myocardial infarction [10], acute hemorrhoidal disease [11], overweight (reduces cardiovascular risk) [12], or type II diabetes [13].

Hes consists of aglycone–hesperetin and a sugar moiety–rutinoside [6]. Its bioavailability is strongly linked to modification by the gut microbiota. Gut microbiota releases the aglycone form—hesperetin. It can be absorbed in the colon and enter the bloodstream or be further modified by the microbiota to yield short-chain fatty acids [14,15].

Due to its low solubility (4.95 μg∙mL^–1^), the bioavailability of Hes is limited, so it is not easy to use as a therapeutic agent, despite its excellent health-promoting potential [16]. Numerous approaches are available and have been reported in the literature to increase the solubility of poorly water-soluble polyphenols. The use of techniques such as freeze-drying [17], spray drying [18], hot-melt extrusion [5], solvent evaporation, and cryomilling [19] has been demonstrated to obtain amorphous solid dispersions of polyphenols. One of the most promising strategies is amorphous solid dispersion. The amorphous form inherently has high energy compared to its crystalline counterpart. For this reason, this state is prone to rapid recrystallization. Usually, a carrier additive is introduced as a stabilizer, which can significantly delay the onset of crystallization [20].

To our knowledge, research on the amorphization of hesperidin can be considered as one of the pioneering processes to improve its solubility. For example, Wei et al. obtained amorphous hesperidin using the wetness impregnation method [21,22], Wdowiak et al. obtained amorphous inclusion complex by solvent evaporation [23]. Nevertheless, other techniques have been widely used to improve the solubility of hesperidin [15]. So far, several methods have allowed us to increase this compound’s solubility and permeability. They were, for example, inclusion complex with cyclodextrin [24], phospholipid complex [25], nanoemulsions [26], nanophytosomes [27], solid dispersion [28,29], and nanodispersion [30]. Hesperidin delivery systems have also been confirmed with chitooligosaccharide by spray drying [31], gold nanoparticles by a chemical synthesis method [32], chitosan nanoparticles [33], and lipid–polymer hybrid nanoparticles by the emulsification solvent evaporation method [34]. For example, Stanisic et al. [26] obtained hesperidin with an approximate purity of 98% from orange pomace by aqueous extraction. They then used hesperidin to formulate nanoemulsions, incorporated them into a cream formulation, and conducted in vitro tests on 3D artificial skin. Majumdar et al. prepared a hesperidin complex with 2-hydroxypropyl-β-cyclodextrin with enhanced solubility. Solutions of the complex were stable at pH 1.2, 3.5, and 7.4 (temperature 25 °C and 40 °C) for two months and unstable at pH 9.0 (temperature 25 °C and 40 °C) after one day (analysis of observed degradation rate constants). These results indicate that hesperidin can undergo alkaline hydrolysis at higher pH and temperature conditions [24].

Our study aimed to obtain amorphous solid dispersions (ASDs) of hesperidin by use of carriers: Soluplus^®^ (Sol), alginate sodium (SA), and hydroxypropylmethylcellulose (HPMC) and to evaluate their solubility in water, phosphate buffer (pH 6.8) and HCl, and apparent solubility in phosphate buffer (pH 6.8) and HCl.

## 2. Results

The obtained amorphous Hes and Hes-carrier systems were analyzed to confirm the amorphous form of Hes. X-ray powder diffraction (XRPD) is useful for identifying the difference between crystalline and amorphous phases. Figure 1a show XRPD diffractograms of crystalline Hes and amorphous Hes obtained by melting.

The crystalline Hes is characterized by a crystal pattern consisting of a series of well-defined sharp peaks at 12.16°, 15.55°, 19.59 °, 21.26°, 22.41°, and 24.80° 2Θ (Figure 1a, black line). The completely amorphous “halo” from the x-ray pattern of melted Hes confirms the formation of a wholly amorphous structure of Hes (Figure 1a, red line). In our previous work, there is a full list of peaks of hesperidin, their relative intensities, Bragg angles 2Θ, and interplanar distances dhkl [23]. Figure 1b–d show XRPD diffractograms of pure crystalline Hes, carriers (Soluplus^®^, alginate sodium, and hydroxypropylmethylcellulose); their physical mixtures (ph. m.); and their binary systems, which were prepared at weight ratios of 1:2 and 1:5.

The diffractograms of Sol, SA, and HPMC are typical of amorphous material. The XRPD patterns of physical mixtures of hesperidin with all carriers at weight ratios of 1:2 and 1:5 corresponded to the superposition of the patterns of individual components. This indicates the lack of formation of a new phase in the solid state. In the case of Hes:Sol, Hes:SA, and Hes:HPMC analyses (ratio 1:2), small crystalline peaks suggested nanoization, while in the case of Hes:Sol, Hes:SA, and Hes:HPMC (ratio 1:5), the absence of Bragg peaks in the diffractogram was indicative of the amorphous nature of the resulting systems.

The obtained systems were then studied by thermogravimetric analysis (TG) and differential scanning calorimetry (DSC). On the basis of TG analysis, the thermal stability of the obtained systems and their physical mixtures were evaluated in relation to Hes and the carrier. The obtained data (unpublished data) allowed appropriate selection of the temperature range during DSC analysis. All thermal degradation profiles were obtained in the same experimental conditions (see Section 4.3.2.).

TG analysis confirmed the degradation of the Hes-carrier systems in the region of the melting point of hesperidin (T_m_ = 260 °C). It is reasonable to perform DSC analysis only to observe the glass transition temperature.

DSC analysis was performed to measure the glass transition temperature (T_g_) of amorphous Hes. Based on the results of the TG analysis, the maximum temperature value for the Hes:carrier systems was determined, which are 215 °C, 190 °C, and 215 °C for Hes:Sol, Hes:SA, and Hes:HPMC systems, respectively. In order to evaporate the water from each formulation before measuring the glass transition temperature (T_g_), all samples were heated on aluminum pans to a temperature of about 100 °C (isotherm: 10 min, see Section 4. Materials and Methods).

For Hes:carrier systems, it was possible to predict their T_g_ value using the Gordon–Taylor (G-T) relationship. This quantitative model uses the T_g_ values of the individual components and predicts the T_g_ of an intermediate dispersion in the T_g_ value of each component. G-T is based on the assumptions that the free volumes of both components are additive and that there are no specific interactions in the system [35]. Based on these assumptions, the resultant T_g_ of binary systems was calculated using Equation (1) (see Section 4.3.3).

Considering the experimental T_g_ value of Hes: 107.1 °C, Soluplus^®^: 79.2 °C, SA: not recorded, HPMC: 134.2 °C (Figure 2), the T_g_ values for Hes:Sol and Hes:HPMC systems were calculated (Table 1).

The predicted T_g_ values were close to the experimental ones. This is in line with the literature, which reports that deviations from ideality are reported in practice [36,37]. This means that the mixing of components is not ideal and is classified as positive and negative deviations (see Appendix A). In our case, only Hes:Sol 1:2 system has a positive deviation. In the case of Hes:Sol and Hes:HPMC systems in a 1:5 ratio, one glass transition temperature confirmed the full miscibility of these amorphous systems.

Fourier-transform infrared spectroscopy attenuated total reflectance (FT-IR-ATR) analyses were performed to identify the main bands of compounds and investigate potential interactions between carriers and Hes. The optimized geometry of the Hes is shown in Appendix A. The quantum chemical calculations of the normal modes of vibrations permit us to assign the bands observed in the experimental spectra to specific vibrations of various functional groups of the molecule.

The most intense bands of the crystalline Hes were observed in the FT-IR-ATR spectra at about 900–1200 cm^–1^, 1500–1700 cm^–1^, and 2850–3550 cm^–1^ (see Appendix A). Comparing the spectrum recorded for the crystalline form with that obtained for the amorphous form indicates the disappearance, shape change, and change in intensity of some characteristic bands (see Figure 3).

In the range 400–900 cm^–1^ we can highlight the bands corresponding to the O–H and C-H vibration. For example, bands located at (crystalline/amorphous Hes, respectively): 554/552 cm^–1^ correspond to the wagging vibration of the O–H bonds at A ring. Bands corresponding to the C-H vibration are located at 816/806 cm^–1^ (at A ring), 849/- cm^–1^ (at A and B ring), 876/868 cm^–1^ (in methylene group at C ring). Next, the range between 900–1200 cm^–1^ is dominated by bands related to the stretching vibration of the C–O bonds in rutinose (910/922 cm^–1^, 970 and 982/984 cm^–1^, 1022/1018 cm^–1^, 1049/1057 cm^–1^). These bands also have additional components related to the stretching vibration of the C–C bonds and the rocking vibration of the O–H and C–H bonds in rutinose. In contrast, the bands corresponding to the rocking vibration of the C–H and stretching vibration of the C–O and C–O–C in hesperedin are mainly located in the range between 1200 and 1500 cm^–1^. The first is located at 1206/- cm^–1^ (in all molecules), 1242/- cm^–1^ (at A ring), 1277/1271 cm^–1^ (at C ring), 1300/1300 cm^–1^ (at A ring), 1327/- cm^–1^ (at B ring), 1341/1339 cm^–1^ and 1356/1381 cm^–1^. The second is located at 1206/- cm^–1^ and 1300/1300 cm^–1^ (in C–O–C between A ring and glucose ring). The last (C–O–C stretching vibration) is located at 1242/- cm^–1^ (in C ring) and 1277/1271 cm^–1^ (at B ring). All bands have additional components (see Appendix A). Bands corresponding to the O–H and C-H rocking vibration are located at 1518/1514 cm^–1^ (O–H r at A ring and C–H r at C ring), 1605/1593 cm^–1^ (O–H r at A ring) and 1645/1634 cm^–1^ (O–H r at A ring). In addition, bands 1605 cm^–1^ and 1645 cm^–1^ have additional components: C=C s in ring A and C=O s at ring C. Between 2850–3550 cm^–1^ are bonds corresponding to the O–H and C–H stretching vibrations. For example, O–H stretching vibrations are located at 3414 cm^–1^ (at A ring), 3476 cm^–1^ and 3543 cm^–1^ (at rhamnose ring) in crystalline Hes. Amorphous Hes has one wide band in this range with a maximum of 3393 cm^–1^. Next, bonds corresponding to the C–H stretching vibrations are located at 2849/2841 cm^–1^, 2940/2934 cm^–1^, 2982/- cm^–1^ (at rhamnose ring), 2918/2914 cm^–1^, and 3080/- cm^–1^ (in methoxy group at B ring).

In the FT-IR-ATR spectra of amorphous hesperidin, in addition to the disappearance of some characteristic bands, the shape of the peaks is visible. They concern the range of 400−800 cm^–1^ (corresponding to deformation of all molecules), 900−1100 cm^–1^ (corresponding to groups at glucose and rhamnose ring), and 1125−1650 cm^–1^ (corresponding to C–O–C, O–H, C–H groups), see Appendix A.

These results confirm that the changes involve the entire structure of hesperidin. Hydrogen bonds are probably responsible for maintaining the amorphous state. This is indicated by changes within the C–H, O–H and C–O–C groups.

The FTIR-ATR spectra for amorphous Hes and the systems of Hes are presented in Figure 4, Figure 5 and Figure 6. In the spectra of the system of Hes:Sol (ratio 1:2 and 1:5), peaks characteristic of Sol predominate (Figure 4, Appendix A) [38,39,40,41].

The changes in the nature of Sol’s bands mainly concern a reduction in intensity. Shifting toward higher wavelengths was observed for the band at 1476 cm^–1^ corresponding to C−O−C s in the ether groups [42]. Nevertheless, bands characteristic of amorphous hesperidin are also visible (743 cm^–1^, 762 cm^–1^, 804 cm^–1^, 1514 cm^–1^, 3383 cm^–1^). These bands in the system’s spectra are shifted and have reduced intensity. Thus, the band corresponding to the deformation of all molecules (743 cm^–1^) in the 1:2 and 1:5 systems is shifted to 745 cm^–1^. Next, the band corresponding to the breathing B ring (762 cm^–1^) in the systems is shifted to 766 cm^–1^. The intense band observable at 804 cm^–1^ (C–H wagging at A ring) shifted toward higher wavelengths by about 4 cm^–1^ (Figure 4a). The band corresponding to the vibrations of the O-H at ring A and C=O at ring C (1514 cm^–1^) does not shift, but almost completely disappears in the spectra of the systems (Figure 4b). The broad band at 3383 cm^−1^ corresponding to the stretching of O-H groups undergoes a slight shift toward higher wavelengths and reduces its intensity (Figure 4c). The observed changes in the systems’ spectra indicate the formation of hydrogen interactions between C–H and/or O–H group of Hes and C–O group of Sol. Stronger intermolecular hydrogen interactions between Hes and Sol seem to be accountable for the complete amorphization of the Hes:Sol 1:5 system, confirmed by XRPD analysis (Figure 1a).

In the spectra of the system of Hes:SA (ratio 1:2 and 1:5), peaks characteristic of SA predominate (Figure 5, Appendix A) [43,44].

Nevertheless, numerous bands characteristic of amorphous hesperidin are also visible (806, 1018, 1057, 1128, 1157, 1179, 1271, 1591, 1634, and 3383 cm^–1^). In all systems, the SA and Hes bands have an altered shape and reduced intensity in relation to the individual component. Again, hydrogen bonds are responsible for maintaining the amorphous state of hesperidin. It is indicated that, in the Hes:SA 1:5 system, stronger interactions between the individual components of the system occurred (including an increase in the intensity of the bands corresponding to the O–H groups at 1026 cm^–1^, C–OH at 1406 cm^–1^ and O–C–O at 1599 cm^–1^ compared to the 1:2 system). This is consistent with XRPD analysis, which showed that full amorphousness was achieved for the 1:5 system (Figure 1b). The observed changes in the systems’ spectra indicate the formation of hydrogen interactions between C–H and/or O–H group of Hes and C–O group of SA.

In the spectra of the system of Hes:HPMC (ratio 1:2 and 1:5), peaks characteristic of amorphous Hes predominate (Figure 6, Appendix A).

Bands corresponding to HPMC [45,46,47,48,49,50,51] do not shift in the spectra of the systems, except for one band observed in the pure carrier at 3460 cm^−1^ (O–H stretching vibration [52]). This band shifts to the lower wavenumber by about 17 cm^−1^. Next, numerous bands characteristic of amorphous hesperidin observed in obtained Hes:HPMC systems have an altered shape, reduced intensity, and are shifted toward higher wavelengths. For example, bands observed at about 760–833 cm^−1^ correspond mainly to molecule deformations and C–H bond vibrations are shifted toward higher wavelengths by about 3 cm^−1^. Then, the bands observed in the range of 1270–1635 cm^−1^, corresponding mainly to vibrations of C–H, O–H, and C–O bonds, are also shifted toward higher wavelengths by about 3 cm^−1^. All this suggests the formation of hydrogen bonds between C–O group of Hes and O–H group of HPMC, which stabilizes the amorphous form of Hes.

It is worth mentioning that ball milling as a method of preparing the amorphous form of Hes did not degrade Hes. In other words, Hes was chemically stable, as confirmed by HPLC analysis. Thus, we avoided a situation in which Hes would have been degraded to its aglycone-hesperetin (see Appendix A).

The solubility of crystalline Hes in water, phosphate buffer (pH 6.8), and HCl 0.1 N were: 0.009 ± 0.001, 0.009 ± 0.002, 0.008 ± 0.003 mg·mL^–1^, respectively. For all obtained Hes systems, an improvement in its solubility was noted compared to pure Hes (Table 2). The solubility of Hes with different carriers in water decreased in the order of Sol > SA > HPMC, and the system of Hes:Sol (ratio 1:5) exhibited the maximum solubility, i.e., 2.710 ± 0.004 mg·mL^–1^. Next, the solubility of Hes with different carriers in HCl decreased in the order of Sol > HPMC > SA, and the system of Hes:Sol (ratio 1:5) exhibited the maximum solubility, i.e., 2.486 ± 0.033 mg·mL^–1^. In contrast, the solubility of Hes with different carriers in phosphate buffer pH 6.8 decreased in the order of Sol > SA > HPMC, and the system of Hes:Sol (ratio 1:5) exhibited the maximum solubility, i.e., 2.616 ± 0.036 mg·mL^−1^.

The obtained amorphous systems impacted the dissolution profile of Hes in phosphate buffer and HCl (Figure 7a and Figure 7b respectively).

The apparent solubility increased compared to the pure compound. Systems with more carriers provided better solubility improvement. Hes systems reach the plateau state in about 60 min. Interestingly, the behavior of alginate sodium depends on pH. In an acidic environment, the dissolved amount of Hes is very similar to the undissolved compound, indicating that the systems cannot stabilize the amorphous state in the dissolution medium.

## 3. Discussion

Hesperidin (Hes) is a flavonoid polyphenol belonging to the flavanone group. It has been found to have significant potential in preventing and treating chronic diseases. However, its further research and use as a therapeutic agent is limited due to its poor bioavailability, which is associated with poor solubility [53,54]. To the best of our knowledge, this work is the first report on preparing amorphous solid dispersions of Hes using ball milling.

XRPD and DSC methods confirmed the amorphous nature of obtained systems, while the FT-IR technique made it possible to determine the molecular interactions between Hes and selected carriers.

The XRPD analysis showed that melting hesperidin and all Hes:carrier systems of 1:5 weight ratio were amorphous. The XRPD patterns of amorphous solid form show a lack of Bragg’s peaks in diffractograms. The appearance of the characteristic “halo effect” on the diffractograms indicates the formation of amorphous Hes. The “halo effect” for amorphous hesperidin delivery systems was also observed by Wei et al. [21,22] and Wdowiak et al. [23].

Morevoer, the TG results confirm that all the obtained systems degrade in the region of the Hes melting point. For this reason, DSC analysis was carried out only to determine the glass transition temperature (T_g_) characteristic of amorphous samples. The analysis of obtained DSC data confirms that the Gordon–Taylor equation can be a suitable tool for predicting T_g_ in amorphous binary systems. One glass transition in hesperidin:Soluplus^®^ (Hes:Sol) and hesperidin:HPMC (Hes:HPMC) 1:5 systems confirmed their complete miscibility. Moreover, the negative deviations suggest weaker Hes–carrier interaction than the Hes–Hes and carrier–carrier interactions. Kanaze et al. confirmed the obtaining of miscible hesperidin–PVP dispersions, which showed one glass transition temperature (T_g_ characteristic for PVP). The observed T_g_ value was lower than that recorded for the pure polymer. Probably low molecular mass compound acted as a plasticizer reducing the T_g_ of PVP and/or some weak interaction was involved between the polymer and the compounds, resulting in the formation of amorphous solid dispersion systems [55]. Yen et al. used infrared spectroscopy, thermal analysis and theoretical prediction of T_g_ values to prove the existence of favorable interactions between tannic acid and polyesters and its miscibility [56]. Lee et al. used the same approach to miscibility analysis and interaction identification of biodegradable polymers with tannic acid composition [57].

Fourier-transform infrared (FT-IR) spectroscopy is a powerful tool for investigating intermolecular API–API and API–polymer interactions. The information on the intermolecular interactions can be detected through the variation of relative spectroscopic vibration bands. FT-IR method is sensitive to hydrogen bond formation and is useful to verify the presence of intermolecular interaction between donor and acceptor groups in different systems. The FT-IR-ATR analysis made it possible to characterize the changes formed in the structure of hesperidin after its amorphization by melting. Comparing the spectrum recorded for the crystalline form with that obtained for the amorphous form indicates fading, shape change and change in intensity of many characteristic bands (Figure 3). The spectrum of amorphous Hes formed the basis for interpreting the changes observed in the spectra of the systems obtained. Numerous bands of amorphous hesperidin were confirmed in all the obtained systems (Figure 4, Figure 5 and Figure 6). Each of the carriers used formed hydrogen bonds with Hes. This is indicated by numerous changes in the shape of the bands corresponding to C–H, O–H, and C–O bonds, among others, a change in their intensity as well as a shift towards higher wavelengths. It is indicated that hydrogen bonds have formed between the C–H and/or O–H group of hesperidin and C–O group of Soluplus^®^, and C–O group of Hes and O–H group of HPMC. The 1:5 systems exhibited stronger hydrogen bonds than the 1:2 systems, which maintained the amorphous state of Hes. The FT-IR analysis combined with the XRPD analysis provided evidence that the stronger bonds present in the 1:5 systems allowed for full amorphization of Hes. The presence of hydrogen bonds responsible for maintaining the amorphous state of hesperidin was confirmed by Wdowiak et al. [23]. In another study Joshi et al. reported no interaction between hesperidin and the polymer in the obtained solid dispersions [28]. Kanaze et al. indicated possible hydrogen bond formation between the hydroxyl groups (O–H) of the compound with carbonyl groups of PVP (>C–O) in solid dispersions of hesperetin–PVP [28].

It is worth mentioning that ball milling as a method of preparing the amorphous form of Hes did not degrade Hes. In other words, Hes was chemically stable, as confirmed by HPLC analysis. Carriers used in amorphous solid dispersions belong to the non-toxic group. Therefore, based on the HPLC analysis, it is suggested that the absence of degradation products indicates obtaining safe hesperidin delivery systems. For example, Soluplus^®^ is a pharmaceutical excipient used primarily in the manufacture of solid dispersions [58,59,60]. Alginate sodium is generally used in pharmaceutical and food industry. It is considered to be non-toxic and biocompatible, and its compatibility with drugs does not affect the pharmacological effects, showing that it has more natural advantages in the sustained and controlled release of drugs [61]. Hydroxypropyl methylcellulose (HPMC) is non-toxic, biocompatible, and biodegradable. It acts as a drug-dispersing and viscosity-modifying agent and is a controlled delivery component in oral formulations [62,63]. More than 14 amorphous solid dispersion-based products that have been approved for marketing consist of HPMC as stabilizer, thus highlighting its importance in the generation of amorphous solid dispersions [64].

One of the most important reasons for the amorphization of active compounds is to improve their solubility. Furthermore, in the case of Hes, studies of its solubility were carried out after obtaining amorphous dispersions with selected carriers.

Hes is characterized by poor solubility (4.95 μg∙mL^–1^) [24]. Compared to pure Hes, the best improvement in solubility was obtained for its amorphous dispersions in Sol when the systems were formulated for the ratio of 1:5. An approximately 300-fold improvement in solubility was found in all media tested (water, HCl 0.1N, and phosphate buffer pH 6.8). The results may be related to this carrier’s possibility of forming micelles. In the case of Soluplus^®^, the solubilization of dissolved Hes by the formation of micelles could contribute to the stabilization of the supersaturation state, which translated into a significant increase in solubility to pure Hes. Moreover, it is noticeable that a greater amount of carrier in the system provided better solubility. In the case of systems of Hes:SA (ratio 1:2 and 1:5), the solubility also improved in each tested media. The best results were obtained in a phosphate buffer solution (pH of 6.8). The system of Hes:SA (ratio 1:2) made it possible to obtain a concentration of 1.605 mg·mL^−1^, which resulted in a 178-fold improvement in solubility over pure hesperidin under these conditions. In turn, the system of Hes:SA (ratio 1:5) in the same conditions led to an even better improvement, reaching a concentration of 1.726 mg·mL^−1^, which translated into a 192-fold increase in solubility. A markedly weaker increase in solubility of Hes:SA systems can be seen in an acidic environment. The increase in the amount of SA in the system was related, in this case, to poorer solubility, in contrast to other media where greater amounts of carrier led to better solubility. This may be because alginate sodium is a pH-dependent carrier. For systems of Hes:HPMC (ratio 1:2 and 1:5), the lowest degree of improvement in solubility was found, i.e., a 79-fold increase for the system of Hes:HPMC (1:5), in phosphate buffer at pH 6.8. For example, Joshi et.al showed an improvement in the solubility of hesperidin in amorphous dispersions—0.328 mg·mL^−1^ in distilled water and 0.362 mg·mL^−1^ in 0.1 N NaOH [28].

In considering the conditions in selected sections of the digestive system, it can be suggested that increasing the solubility, particularly in the colon region, may help to improve the bioavailability of Hes. Increasing the amount of dissolved hesperidin would enable intestinal bacteria to convert it into Hesperetin, which could then be absorbed into systemic circulation and exert a pharmacological effect [65].

For the hesperidin systems, the dissolution rate of the drug in phosphate buffer improved significantly and increased from about 50% at 30 min to about 60–70% at 120 min. The apparent solubility of Hes with different carriers in phosphate buffer decreased in the order of Hes:Sol 1:5 > Hes:HPMC 1:5 > Hes:SA 1:5 > Hes:Sol 1:2 > Hes:SA 1:2 > Hes:HPMC 1:2. However, in HCl, the best solubility improvement was confirmed for Hes:Sol 1:5, Hes:Sol 1:2 and Hes:HPMC 1:5 and increased from about 45–55% at 30 min to about 57–63% at 60 min. Kanaze et. al. confirmed that the presence of crystalline hesperidin in PVP dispersions did not improve the release profiles [55]. Dispersions of hesperidin obtained by Joshi et al. had a dissolution rate between 20 and 80% in 0.1 N NaOH [28].

Physical stability studies in amorphous systems give information about the lifetime of an amorphous state, which is related to improved physicochemical properties. Stability studies carried out at the point (0 and 7 days, XRPD analysis) for two stress conditions (25 °C/RH = 76.4% and 60 °C/RH = 76.4%) confirmed the poor stability of all systems (Appendix A) [66]. Despite the appearance of crystal peaks in diffractograms, it can be noted that in the case of the 1:5 systems, the peaks were smaller than in the 1:2 systems, thus suggesting that the amount of polymer in the dispersion may affect the stability of the amorphous form. The conducted research is a preliminary assessment, and to draw more detailed conclusions about stability, more in-depth tests should be carried out under different humidity and temperature conditions due to the significant influence of the above-mentioned factors on the phenomenon of crystallization in amorphous solid dispersions [67].

The problem of limited stability may be solved by obtaining triple systems consisting of a mixture of polymers and an active substance. For example, Alshahrani et al. [68] prepared an amorphous dispersion of carbamazepine using a combination of Soluplus^®^ to act as a solubilizer and hydroxypropyl methylcellulose acetate succinate (HPMCAS) as an amorphous stabilizer. The obtained systems were characterized by a significant improvement in the dissolution rate. In addition, they ensured the stability of the amorphous form for over 12 months in conditions of increased temperature and humidity. In addition, Prasad et al. [69] obtained amorphous dispersions of indapamide with Eudragit^®^ and PVP K90. In stability studies, the triple dispersion of indapamide-PVP K90-Eudragit^®^ performed much better, assuring the system’s stability in conditions of increased temperature and humidity up to 180 days. In comparison, the dual indapamide–polymer systems gave stability for up to 30 days.

## 4. Materials and Methods

### 4.1. The Active Substance and Materials

Hes was supplied by LKT Laboratories (St. Paul, MN, USA). From Sigma Aldrich (St. Louis, MO, USA) the following were obtained: alginate sodium (SA), hydroxypropylmethylcellulose (HPMC). POCH (Gliwice, Poland) provided: acetic acid (98–100%) and sodium chloride. Soluplus^®^ (polyvinyl caprolactam-polyvinyl acetate-polyethylene glycol graft copolymer), an analytical weighed amount of HCl, 1 N, and sodium dihydrogen phosphate were supplied by BASF SE (Ludwigshafen, Germany), Chempur (Piekary Slaskie, Poland), and PanReac AppliChem ITW Reagents (Darmstadt, Germany), respectively. Methanol of HPLC grade was supplied by J. T. Baker (Center Valley, PA, USA). High-quality pure water was prepared using a Direct-Q 3 UV purification system (Millipore, Molsheim, France, model Exil SA 67120).

### 4.2. Preparation of Amorphous Hesperidin and Hesperidin-Carrier Amorphous Solid Dispersions

Amorphous hesperidin was prepared by melting 17 mg of crystalline sample of Hes in an open aluminum DSC pan because the grinding method failed to obtain amorphous Hes. For this purpose, a differential scanning calorimeter DSC 214 Polyma (Netzsch, Selb, Germany) was used. Heating–cooling parameters were: 25–100 °C (↑, 10 K min^–1^), 100 °C (→, 10 min), 100–255 °C (↑, 10 K min^–1^), 255–25 °C (↓, 40 K min^–1^), where ↑ and ↓ are the dynamic mode, and → is the isothermal mode. The obtained sample was grated in a mortar and subjected to XRPD and FT-IR analysis.

Preparation of amorphous solid dispersions of Hes was carried out with various carriers: Soluplus^®^, alginate sodium, and hydroxypropylmethylcellulose. Milling was used as an amorphization technique. The mixtures of the Hes and carrier in 1:2 and 1:5 weight ratios (100 mg:200 mg and 50 mg:250 mg) were added to a 5 mL glass tube and mixed using a vortex mixer for 30 s to obtain the physical mixture. A total of 300 mg of each mixture was transferred to a 5 mL stainless-steel jar that would fit the MIXER MILL MM 400 (Retsch, Haan, Germany) and that contained one ball (stainless steel, 7 mm diameter). Then, the milling time was set at 12 min, and the milling frequency was 30 Hz. The total grinding time was 60 min (5 cycles of 12 min). A break of 7 min was applied between the cycles to avoid overheating and melting the sample. The process was performed at room temperature. The obtained systems appeared as a homogeneous, fine powder.

### 4.3. The Characterization of Hesperidin-Carrier Amorphous Solid Dispersions

#### 4.3.1. X-ray Powder Diffraction (XRPD)

The characterization of the ASD of Hes was carried out by the XRPD method. Diffraction patterns were recorded on a Bruker D2 Phaser diffractometer (Bruker, Germany) with CuKα radiation (1.54060 Å) at a tube voltage of 30 kV and a tube current of 10 mA. The angular range was 5° to 45° with a step size of 0.02° and a counting rate of 2 s·step^−1^. The acquired data were analyzed using Origin 2021b software (OriginLab Corporation, Northampton, MA, USA).

#### 4.3.2. Thermogravimetric Analysis (TG)

Thermogravimetric (TG) analysis was performed using TG 209 F3 Tarsus^®^ micro-thermobalance (Netzsch, Selb, Germany). Then, about 6 mg powdered samples were placed in Al_2_O_3_ 85 µl, open, and heated at a scanning rate of 10 K·min^−1^ from 25 to 400 °C in a nitrogen atmosphere with a flow rate of 250 mL·min^−1^. The obtained TG data were analyzed using the computer program Proteus 8.0 (Netzsch, Selb, Germany). The obtained TG data were analyzed using the Proteus 8.0 software (Netzsch, Selb, Germany).

#### 4.3.3. Differential Scanning Calorimetry (DSC) and Gordon–Taylor equation

Thermal analysis was performed using DSC 214 Polyma differential scanning calorimeter (Netzsch, Selb, Germany). DSC was used to study all samples’ glass transition temperature (T_g_). A blank aluminum DSC pan was used as the reference sample, powdered samples of 9–10 mg were placed in sealed pans with a hole in the lid, and different sample heating modes were used to observe the T_g_ (see Appendix A). A nitrogen atmosphere was used with a flow rate of 250 mL·min^−1^. The obtained DSC data were analyzed using the Proteus 8.0 software (Netzsch, Selb, Germany). The results were visualized using the Origin 2021b software (OriginLab Corporation, Northampton, MA, USA). The theoretical glass transition temperatures were calculated using the Gordon–Taylor equation:(1)$$T_{g}=\frac{w_{1}T_{g1}+K_{GT}w_{2}T_{g2}}{w_{1}+K_{GT}w_{2}}$$

$w_{1},\;w_{2}$—weight fraction of hesperidin and carrier, respectively. $T_{g},\;T_{g1},\;T_{g2}$—predicted glass transition temperature of a binary system; glass transition temperature of hesperidin, and glass transition temperature of the carrier, respectively. $K_{GT}$—constant indicates a measure of interaction between two components. $K_{GT}$ expressed mathematically:(2)$$K_{GT}=\frac{\rho_{1}T_{g1}}{\rho_{2}T_{g2}}$$
where $\rho_{1},\;\rho_{2}$—the densities of two components (Hes: 1.7 g·cm^−3^, Sol: 1.082 g·cm^−3^, HPMC: 1.39 g·cm^−3^), $T_{g1},\;T_{g2}$—glass transition temperature of hesperidin and carrier, respectively.

#### 4.3.4. Fourier-Transform Infrared Spectroscopy (FTIR) and Density Functional Theory (DFT) Calculations

The FTIR-ATR spectra were collected on an IRTracer-100 spectrophotometer. All spectra were measured between 400 and 4000 cm^−1^ in the absorbance mode. The following spectrometer parameters were used: resolution 4 cm^−1^, number of scans 400, apodization Happ–Genzel. The sample was placed directly on the ATR crystal. Solid samples were pressed against the ATR crystal, and the ATR-FTIR spectrum was scanned. All infrared spectra were acquired and further processed (baseline correction, normalize) with LabSolution IR software (version 1.86 SP2, Shimadzu, Kyoto, Japan). The results were interpreted by comparing the FTIR peaks of pure samples with those of prepared systems.

The molecular geometries of Hes were optimized using the Density Functional Theory (DFT) method with Becke’s three-parameter hybrid functional (B3LYP) implemented with the standard 6–311G(d,p) as a basis set. The calculations of normal mode frequencies and intensities were also performed. Applicated to PL-Grid platform (website: www.plgrid.pl, accessed on 11 March 2021) equipped with Gaussian 09 package (Wallingford, CT USA) for DFT calculation. The GaussView (Wallingford, CT USA, Version E01) program was used to propose an initial geometry of investigated molecules and to visually inspect the normal modes. The obtained data were analyzed using the Origin 2021b software (OriginLab Corporation, Northampton, MA, USA).

### 4.4. Solubility and Apparent Solubility Studies of Hesperidin and Its Amorphous Solid Dispersions with Carriers

#### 4.4.1. HPLC Method

Concentrations of Hes during apparent solubility studies of the pure compound and its amorphous solid dispersions with carriers were measured by high-performance liquid chromatography with the DAD detector (HPLC-DAD). In this study, Shimadzu Nexera (Shimadzu Corp., Kyoto, Japan) equipped with an SCL-40 system controller; DGU-403 degassing unit; LC-40B XR solvent delivery module; SIL-40C XR autosampler; CTO-40C column oven; and SPD-M40 photodiode array detector were used. For the stationary phase, a Dr. Maisch ReproSil-Pur Basic-C18 100 Å column, 5 µm particle size, 250 × 4.60 mm (Dr. Maisch, Ammerbuch-Entringen, Germany), was used. The mobile phase was methanol/0.1% acetic acid (70:30 *v/v*). The mobile phase was vacuum filtered through a 0.45 µm nylon filter (Phenomenex, CA, USA). The experimental conditions were as follows: 1.0 mL·min^−1^ flow rate, wavelength 284 nm, and the column temperature was set at 30 °C. The injection volume was 10 µL for both solubility and dissolution rate studies. The retention time of Hes was 3.2 min.

#### 4.4.2. Media for Solubility and Apparent Solubility Studies

High-quality pure water was prepared using a Direct-Q 3 UV purification system (Millipore, Molsheim, France, model Exil SA 67120). Phosphate buffer at pH 6.8 was prepared according to the following description: in a 1000 mL volumetric flask, we placed 250 mL of 0.2 N potassium dihydrogen phosphate solution, then added 112 mL of 0.2 N sodium hydroxide solution and filled the mixture up to 1000 mL with distilled water. A total of 0.1 M HCl was obtained from the analytical weighed amount in accordance with the BASF SE (Ludwigshafen, Germany) recommendations.

#### 4.4.3. Procedure of Solubility Studies

An excess amount of the obtained systems was placed in a 10 mL glass tube; then, 5.0 mL of medium (water, phosphate buffer (pH 6.8) or HCl 0.1 N) was added. The samples were placed in a stirring device (rotated at 300 rpm) at room temperature for 3 h. The obtained solutions were filtered through a 0.45 μm nylon membrane filter (Sigma-Aldrich, St. Louis, MO, USA) and analyzed for Hes by means of HPLC analysis. The study was performed in triplicate.

#### 4.4.4. Procedure of Apparent Solubility Studies

The apparent solubility study was performed in the paddle apparatus. The equivalent of 5 mg of Hes and systems were placed in the paddle apparatus. The study was carried out in HCl 0.1 N and phosphate buffer pH 6.8. The vessels were filled with 500 mL of medium, the temperature was maintained at 37 °C, and the paddles were set at a stirring speed of 50 rotations per minute. The 2.0 mL samples were withdrawn at predetermined time points with the replacement of equal volumes of temperature-equilibrated media and filtered through a membrane filter (0.45 μm).

## 5. Conclusions

Our studies confirmed obtaining amorphous dispersions of hesperidin with Soluplus^®^, sodium alginate, and HPMC in a 1:5 *w/w* ratio. Thermal analysis indicates that the Gordon–Taylor equation could be suitable for predicting glass transition values and interaction forces in binary systems of hesperidin. FT-IR analysis confirmed the formation of weak hydrogen bonds between hesperidin and carrier. The amorphization of hesperidin allowed for a significant increase in its solubility and apparent solubility. All obtained amorphous solid dispersions are promising delivery systems of hesperidin, and they can pose a more effective approach to treating civilization diseases.

## Figures and Tables

**Figure 1 ijms-23-15198-f001:**
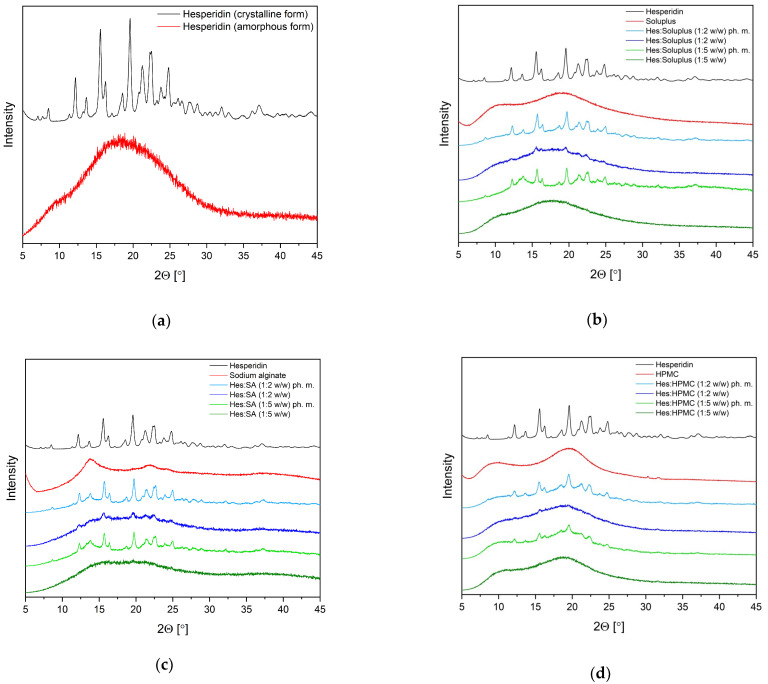
The X-ray powder diffraction analysis: (**a**) Hesperidin crystalline form and Hesperidin amorphous form; (**b**) Hesperidin (Hes), Soluplus^®^ (Sol), Hes:Sol 1:2 *w/w* ph. m., Hes:Sol 1:2 *w/w* system, Hes:Sol 1:5 *w/w* ph. m., Hes:Sol 1:5 *w/w* system; (**c**) Hesperidin, alginate sodium (SA), Hes:SA 1:2 *w/w* ph. m., Hes:SA 1:2 *w/w* system, Hes:SA 1:5 *w/w* ph. m., Hes:SA 1:5 *w/w* system; (**d**) Hesperidin, hydroxypropyl methylcellulose (HPMC), Hes:HPMC 1:2 *w/w* ph. m., Hes:HPMC 1:2 *w/w* system, Hes:HPMC 1:5 *w/w* ph. m., Hes:HPMC 1:5 *w/w* system.

**Figure 2 ijms-23-15198-f002:**
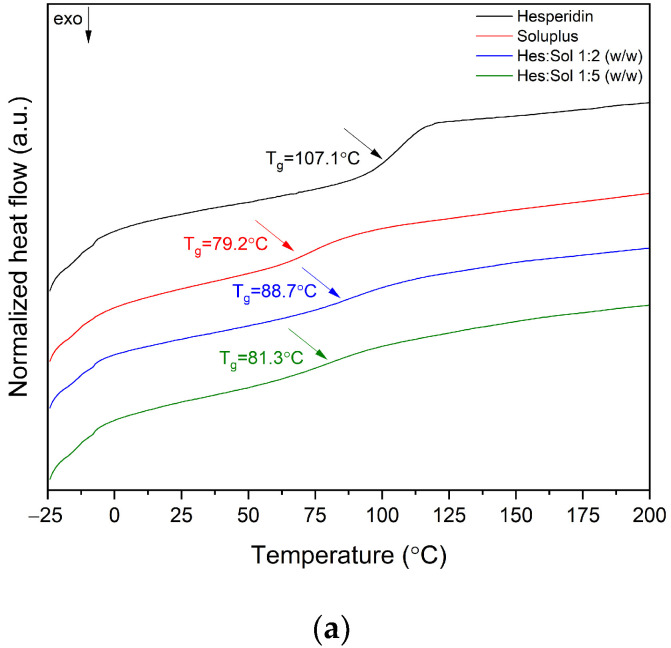
The differential scanning calorimetry (DSC) analysis-second heating: (**a**) Hesperidin (Hes), Soluplus^®^ (Sol), Hes:Sol 1:2 *w/w* system, Hes:Sol 1:5 *w/w* system; (**b**) Hesperidin, alginate sodium (SA), Hes:SA 1:2 *w/w* system, Hes:SA 1:5 *w/w* system; (**c**) Hesperidin, hydroxypropylmethylcellulose (HPMC), Hes:HPMC 1:2 *w/w* system, Hes:HPMC 1:5 *w/w* system.

**Figure 3 ijms-23-15198-f003:**
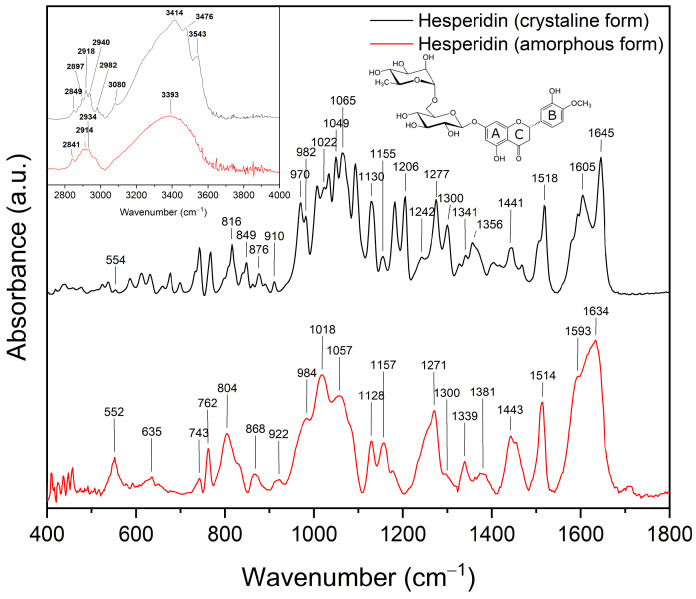
The FT-IR spectra: hesperidin crystalline form (black) and hesperidin amorphous form (red). Figure shows the structure of hesperidin with the positions of rings A, B and C marked.

**Figure 4 ijms-23-15198-f004:**
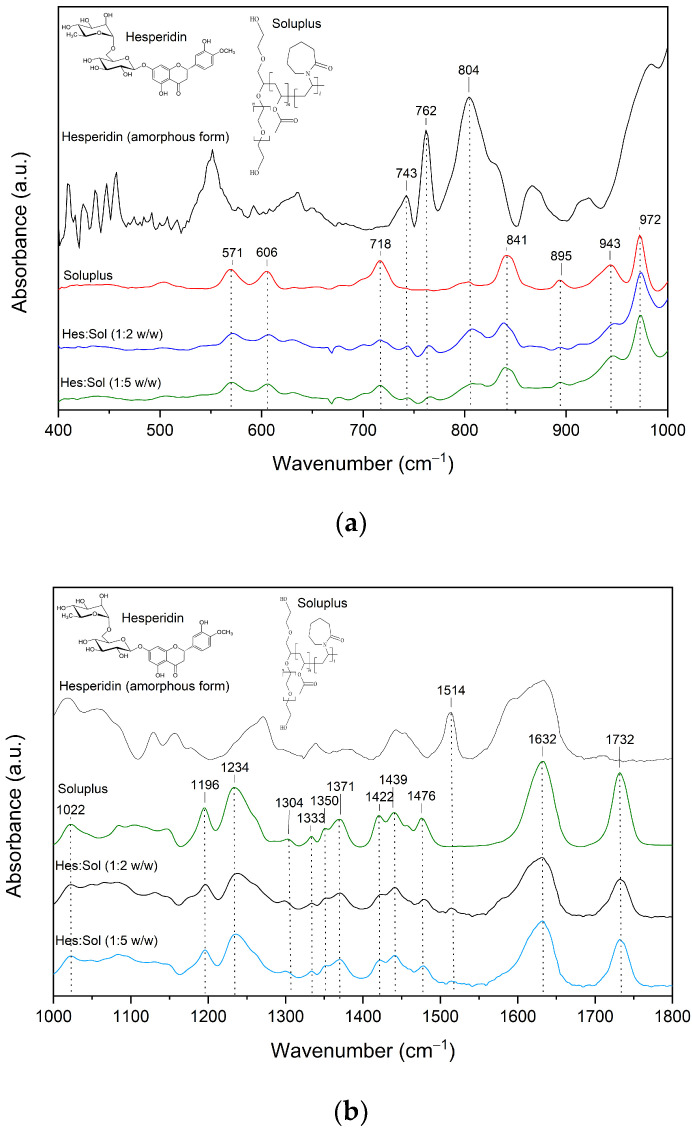
The FT-IR analysis-Hesperidin (amorphous form), Soluplus^®^, Hes:Sol 1:2, Hes:Sol 1:5; (**a**) range 400–1000 cm^−1^, (**b**) 1000–1800 cm^−1^, (**c**) range 2600–4000 cm^−1^.

**Figure 5 ijms-23-15198-f005:**
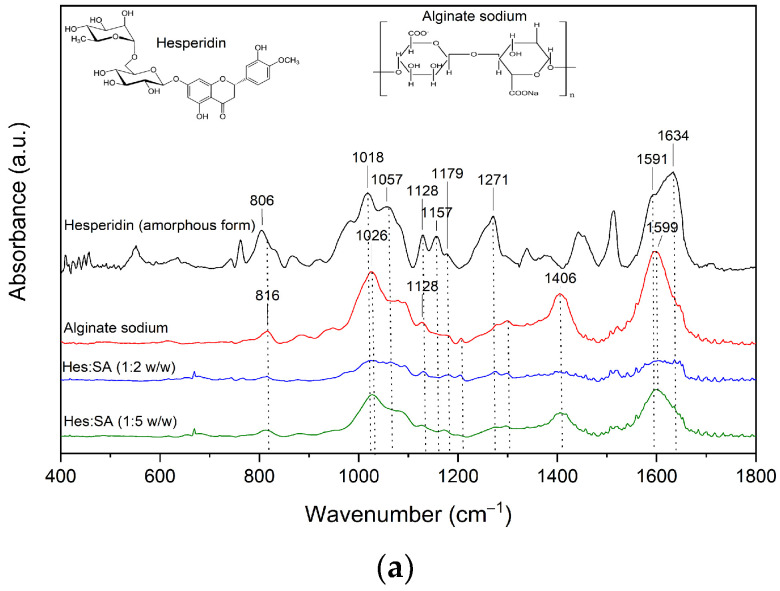
The FT-IR analysis—Hesperidin (amorphous form), Alginate sodium, Hes:SA 1:2, Hes:SA 1:5; (**a**) range 400–1800 cm^−1^, (**b**) range 2600–4000 cm^−1^.

**Figure 6 ijms-23-15198-f006:**
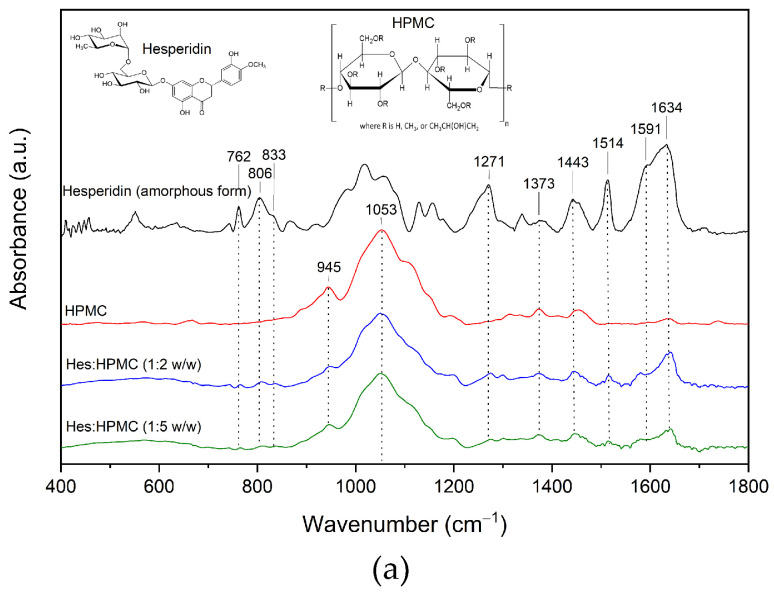
The FT-IR analysis—Hesperidin (amorphous form), HPMC, Hes:HPMC 1:2, Hes:HPMC 1:5; (**a**) range 400–1800 cm^−1^, (**b**) range 2600–4000 cm^−1^.

**Figure 7 ijms-23-15198-f007:**
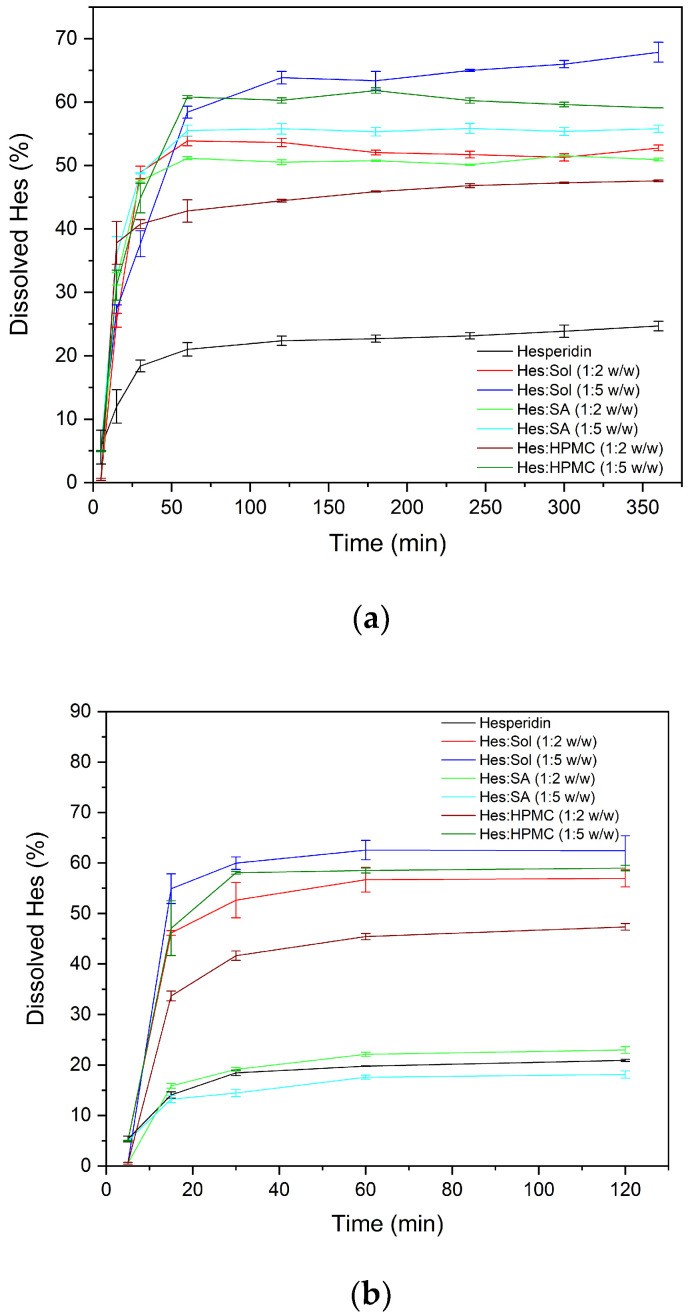
The dissolution profiles of Hes systems’ in (**a**) phosphate buffer and (**b**) HCl.

**Table 1 ijms-23-15198-t001:** Summary of the most important parameters of thermal analysis of Hes: carrier systems.

	Mass (mg)	ΔCp (J·(g K)^−1^)	T_g, calc_ (°C)	T_g, exp_ (°C)	Deviation
Hes	10.24	0.431	−	107.10 ± 0.65	−
Sol	9.45	0.247	−	79.20 ± 0.55	−
Hes:Sol 1:2	9.28	0.192	84.52 ± 0.58	88.70 ± 0.86	+
Hes:Sol 1:5	8.79	0.171	81.60 ± 0.56	81.30 ± 0.36	*
HPMC	6.43	0.211	−	134.20 ± 0.93	−
Hes:HPMC 1:2	10.74	0.105	125.02 ± 0.33	114.30 ± 0.67	−
Hes:HPMC 1:5	10.23	0.239	129.59 ± 0.61	122.10 ± 0.76	−

T_g, calc_—glass transition temperature (calculated), T_g, exp_—glass transition temperature (experimental), positive deviation (+), negative deviation (−), the deviation cannot be determined (*).

**Table 2 ijms-23-15198-t002:** Solubility of hesperidin in amorphous dispersion with carriers.

Medium	System	Concentration [mg mL^−1^]	Improved Solubility [–Fold]
Water	Hesperidin	0.009 ± 0.001	–
	Hes:Sol 1:2	1.149 ± 0.011	128
	Hes:Sol 1:5	2.710 ± 0.004	301
	Hes:SA 1:2	1.059 ± 0.008	118
	Hes:SA 1:5	1.184 ± 0.003	132
	Hes:HPMC 1:2	0.088 ± 0.001	10
	Hes:HPMC 1:5	0.622 ± 0.002	69
phosphate buffer (pH 6.8)	Hesperidin	0.009 ± 0.002	–
	Hes:Sol 1:2	0.869 ± 0.020	97
	Hes:Sol 1:5	2.616 ± 0.036	291
	Hes:SA 1:2	1.605 ± 0.014	178
	Hes:SA 1:5	1.726 ± 0.003	192
	Hes:HPMC 1:2	0.068 ± 0.001	8
	Hes:HPMC 1:5	0.711 ± 0.007	79
HCl 0.1 N	Hesperidin	0.008 ± 0.003	–
	Hes:Sol 1:2	0.777 ± 0.009	97
	Hes:Sol 1:5	2.486 ± 0.033	311
	Hes:SA 1:2	0.055 ± 0.002	7
	Hes:SA 1:5	0.027 ± 0.002	4
	Hes:HPMC 1:2	0.064 ± 0.001	8
	Hes:HPMC 1:5	0.075 ± 0.001	9

## Data Availability

The data are contained within the article and Appendix A.

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
