# Peer review of "Amorphous Solid Dispersion of Hesperidin with Polymer Excipients for Enhanced Apparent Solubility as a More Effective Approach to the Treatment of Civilization Diseases"

_ijms, 2022, doi:10.3390/ijms232315198_

Round 1
Reviewer 1 Report
Title :
Line 3-4. Please make sure the authors use uppercase and lowercase letters appropriately in the title.
Abstract : The Abstract needs to be rewritten.
Line 12-15. This part is suitable for the introductory section. It is better to write more concisely. Please check the unit display.
Line 18-25. Only test methods or items were listed in this part. It is necessary to summarize the experimental results and their implications in Abstract.
Main Text :
Line 56-58. The wording of this sentence is too subjective. There are many solubilization/dispersion processes and technologies. Amorphization can be considered as “one of the pioneering” processes. Suggest more solubilization/dispersion processes and references.
Figure 4. It is not the structure of Soluplus. Please check.
Line 297-299. Solid dispersion of hesperidin has been reported in many studies. (For example: J Appl Polym Sci 102: 460–471, 2006. https://doi.org/10.33263/LIANBS122.050 ) Compare the results of this article with other studies of hesperidin dispersion.
Line 308. Please check if it is grammatically correct.
Line 317. Please check if it is grammatically correct.
Line 337-345. Please check the sentences are correct.
And see the “Instructions for Authors (Abbreviation section)” and check the full text.
Line 340. Explain about Soluplus.
Reviewer 2 Report
1. The solubility of HSP should be tested by experimental results, which is important to support this conclusion and display solubility enhancement in the title.
2. Toxicity study should be performed.
3. Discussion should be elaborated for different analytical techniques perfomed.
4. Stability study is missing.
5. Figures have not the suitable quality, please improve them.
6. The conclusion section should be extended according to the main results.
7. Overall, the English style needs improving.
Round 2
Reviewer 2 Report
I can accept it.